# Real-world management, resource use, patient-reported outcomes and adherence in patients receiving direct oral anticoagulants for first stroke attributed to non-valvular atrial fibrillation in secondary care: A UK mixed-methods observational study

James Uprichard[1]*, Liqun Zhang[1], Anand Dixit[2], Yaqoob Bhat[3], Amit Mistri[4], Dipankar Dutta[5], Khalid Rashed[6], Dumin Karunatilake[7], Chris Hatton[8], Joe Eva[9], Amelia Reed[8]

**1** St George's University Hospitals NHS Foundation Trust, Blackshaw Road, Tooting, London, United Kingdom, **2** Newcastle upon Tyne Hospitals NHS Foundation Trust, Freeman Hospital, High Heaton, Newcastle Upon Tyne, United Kingdom, **3** Aneurin Bevan University Health Board, St Cadoc's Hospital, Lodge Road, Caerleon, Newport, United Kingdom, **4** University Hospitals of Leicester NHS Trust, Infirmary Square, Leicester, United Kingdom, **5** Gloucestershire Hospitals NHS Foundation Trust, Great Western Road, Gloucester, Gloucestershire, United Kingdom, **6** Yeovil District Hospital NHS Foundation Trust, Higher Kingston, Yeovil, Somerset, United Kingdom, **7** Taunton and Somerset NHS Foundation Trust, Taunton, Somerset, United Kingdom, **8** Medical Department, Daiichi Sankyo U.K. Ltd., Uxbridge Business Park, Sanderson Road, Uxbridge, United Kingdom, **9** OPEN Health, The Weighbridge, Brewery Courtyard, Marlow, United Kingdom

* j.uprichard@nhs.net

## Abstract

This real-world study investigated the patient-related factors, characteristics, and outcomes of adult patients with non-valvular atrial fibrillation (NVAF) receiving a direct oral anticoagulant (DOAC) for secondary stroke prevention. This was a multi-centre, mixed-methods, non-interventional study conducted in 8 UK secondary care National Health Service centres. The study included adult patients who presented with first ischaemic stroke associated with NVAF without previous anticoagulants. Group 1 included all patients. Group 2 is comprised of prospectively enrolled patients who were initiated on apixaban (n = 49), edoxaban (n = 39) or rivaroxaban (n = 5) post-first stroke from Group 1. The primary objective (Group 1) was to describe patients' demographics, clinical characteristics, and medical history, stratified by the anticoagulant prescribed. The secondary objectives (Group 2) were to describe the patient management pathways, hospital resource use and clinical assessments associated with DOAC treatment, and the patient-reported satisfaction and experience of DOAC treatment. 234 patients were recruited from 8 centres (Group 1). Baseline $CHA_2DS_2$-VASc risk scores ranged from 2–7; 70% (157/224) had a score of ≥4. 86% (n = 202/234) of patients presented with stroke at accident and emergency. For Group 2, the median time from stroke to first DOAC dose was 6 (IQR, 2.0–10.2; n = 88)

**Data availability statement:** Data cannot be shared publicly. Data are available from Daiichi Sankyo UK at medinfo@daiichi-sankyo.co.uk for researchers upon reasonable request.

**Funding:** This work was funded by Daiichi Sankyo UK Ltd, who were involved in the design of the study and the interpretation and reporting of the results. Daiichi Sankyo provided funding for and were involved in the drafting and revision of this manuscript. Medical writing support was provided by Will Cottam, PhD, of OPEN Health , London (UK), funded by Daiichi Sankyo UK Ltd.

**Competing interests:** JU reports having received support for the present manuscript including sponsorship of the study and medical writing support from Daiichi Sankyo. AM reports having received payment or honoraria from Daiichi Sankyo; support for attending meetings and/or travel from Daiichi Sankyo. CH and AR are employees of Daiichi Sankyo UK. JE was an employee of OPEN Health, London for the duration of the study. KR, LZ, AD, YB, DD, and DK report no conflicts of interest to declare.

days; 50% patients had ≥ 1 outpatient visit recorded related to AF or DOACs. At 3 and 6 months, 73% (46/63) and 83% (43/52) had high (score of 8) Morisky Medication Adherence Scale score (MMAS-8), respectively. No patients reported being dissatisfied at 3 or 6 months post-DOAC initiation. The study findings demonstrate high levels of adherence, persistence, and treatment satisfaction in the 6 months post-initiation of DOAC after first stroke attributable to NVAF in patients. The presented results provide clinicians with valuable insights into the experience of post-stroke patients with NVAF receiving treatment with a DOAC for secondary prevention of stroke during the 6 months post-stroke.

## Introduction

Atrial fibrillation (AF) is the most common cardiac arrhythmia worldwide and is a common cause of ischaemic stroke [1–3]. The prevalence of AF is rising in line with the ageing population, and it has been estimated that approximately 20% of strokes occurring each year in the United Kingdom (UK) are related to AF [4–6]. Subsequently, it is important that AF is diagnosed in good time and effectively treated.

The risk of stroke is increased by fivefold in patients with non-valvular AF (NVAF), yet this risk can be reduced by receiving anticoagulation therapy [7]. The National Institute for Health and Care Excellence (NICE) recommend that anticoagulation therapy is considered for patients at risk of stroke, as assessed by the $CHA_2DS_2$-VASc score. Specifically, it is recommended that patients with $CHA_2DS_2$-VASc score of ≥2 are offered a direct oral anticoagulant (DOAC) to prevent the risk of stroke [8,9].

DOACs currently recommended in the UK include dabigatran, apixaban, rivaroxaban and edoxaban [8,10–13]. Whilst dabigatran, the first DOAC to market, acts as a direct thrombin inhibitor; apixaban, edoxaban and rivaroxaban act via direct inhibition of the activated factor Xa (FXa-inhibiting DOACs). Following the publication of NICE guidelines for each of the DOACs [10–13], the use of DOACs has significantly increased in recent years [14–16]. Although NICE treatment guidelines are available, real-world usage may not reflect this, with approximately 1 in 5 patients receiving incorrect DOAC dosage in a recent study [17]. Publication of the new National Health Service England (NHSE) DOAC commissioning recommendations indicate how the landscape of this therapy area has changed and that DOAC use is becoming more widespread [18]. The increasing use of DOACs over vitamin K antagonists is likely reflective of their relative benefits including their ease of administration (due to the use of fixed doses), an improved efficacy:safety ratio [19], a favourable bleeding profile (although the risk of gastrointestinal bleeding is higher than that observed for warfarin [20]), fewer drug interactions [21] and utilisation without the need for routine monitoring.

Real-world use and effectiveness of FXa-inhibiting DOACs has been described previously in patients with AF in routine clinical practice [22–25], however, there is limited information regarding the impact of patient-related factors, including education and adherence, that may also influence treatment effectiveness. An increased

understanding of such patient-related factors, characteristics, and outcomes of patients with AF receiving a DOAC for secondary stroke prevention would help improve the treatment decision making for clinicians and patients alike. There is currently little known about the patient's perspective on treatment factors, particularly in patients with AF receiving FXa-inhibiting DOAC therapy for secondary stroke prevention.

To address this knowledge gap, this UK mixed-methods real-world study investigated the patient characteristics, management pathways and patient-reported outcomes related to the use of three FXa-inhibiting DOACs in clinical practice in patients following first stoke associated with NVAF: apixaban, edoxaban and rivaroxaban [26].

## Methods

### Study design

This was a UK, multi-centre, mixed-methods, non-interventional study using retrospective data from medical records and prospective data collection from medical records and questionnaires (for patient-reported data). Full details of the study design and methodology are published elsewhere [26]. The study was conducted across 8 UK secondary care National Health Service (NHS) centres.

The study recruited two overlapping groups of patients. The target sample size for Group 1 was 300 patients who presented with first stroke associated with NVAF (the index event). All patients were eligible for inclusion in the study if they presented to the study centre with a first ischaemic stroke which was, in the clinician's opinion, attributable to NVAF and were aged ≥18 years at the time of first stroke. In addition, to have been eligible for Group 2, patients in Group 1 must have been initiated on apixaban, edoxaban or rivaroxaban after their first stroke. All patients were excluded from participation in the study if they were unwilling or unable to give written informed consent, if their medical records were not available for review, if they had been prescribed anticoagulants for any purpose in the 12 months prior to the stroke diagnosis, if they had prior or current haemorrhagic stroke, if they had a diagnosis of transient ischaemic attack, and/or if they had a severe cognitive or emotional deficit. Patients from Group 1 were excluded from Group 2 if they were unwilling or unable to complete the patient-reported questionnaires. The study thus includes patients with known AF prior to stroke who were not anticoagulated in addition to those presenting with stroke who were found to have AF at the time of presentation or after. Data collected for patients in Group 1 were retrospectively collected from medical records up to 12 months prior to, and up to one month following the index event (see **Fig 1**). Group 2 included the first 50 eligible and consenting patients initiated on apixaban, edoxaban or rivaroxaban post-index from Group 1, with a target sample size of 50 patients per treatment (n = 150 total). Data for Group 2 were collected prospectively up to 7 months (6 months + 1 month window for the completion of outstanding questionnaires) after the initiation of DOAC. Patients were contacted and recruited by members of their direct care team between 23 January 2019 and 30 June 2021; recruitment was paused between April 2020 to June 2020 due to the COVID-19 pandemic. All patients gave informed consent to take part in the study. The study design and eligibility criteria for Group 1 and Group 2 are summarised in **Fig 1**. Approvals from the London - South East Research Ethics Committee (REC) and Health and Care Research Wales (HCRW), were obtained prior to study initiation (REC reference: 18/LO/1923; IRAS project ID: 253458). Written consent was obtained from study participants following study approval and initiation. Study authors that were involved in data collection had access to information that could identify participants as they were members of their direct care team (JU/LZ/AD/YB/AM/DD/LR/DK), authors not involved in data collection were only able to access pseudonymised data (CH/JE/AR).

### Study objectives, outcomes and data collection

The primary objective of the study was to describe the demographics, clinical characteristics and medical history of adult patients presenting with a first ischaemic stroke related to NVAF who had not received anticoagulants in the 12 months prior to the stroke, stratified by the anticoagulant treatment received for secondary stroke prevention. This primary

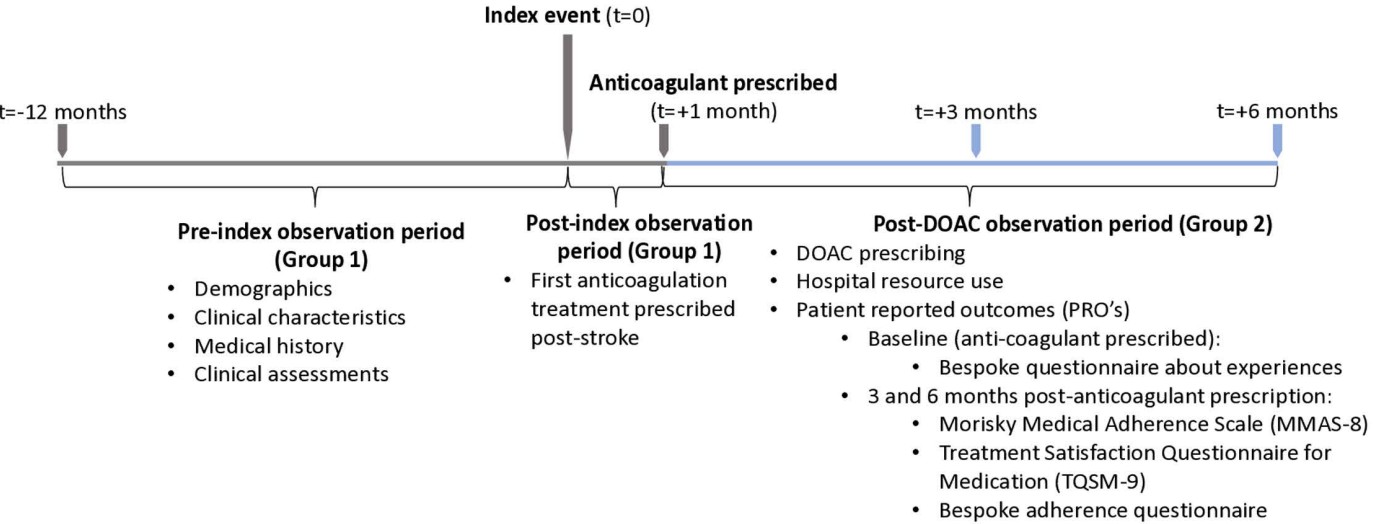

**Group 1 inclusion criteria:**
- Patients presenting with a first ischaemic stroke which is (in the clinician's opinion) attributable to non-valvular atrial fibrillation
- Patient aged ≥18 years at time of first stroke

**Patients were ineligible for inclusion in Group 1:**
- If they had been prescribed anticoagulants in the 12 months prior to the stroke diagnosis (for any indication)
- If they had a history of transient ischaemic attack or haemorrhagic stroke, or had a severe cognitive or emotive deficit at the time of consent.
- If their medical records were not available
- If they would not provide informed consent for researchers to access their medical records

**Group 2 inclusion criteria:**
**Patients from Group 1 could also be included in Group 2:**
- If they had been initiated on apixaban, edoxaban or rivaroxaban for the secondary prevention of stroke.

**Patients from Group 1 were ineligible to be included in Group 2:**
- If they did not want, or were unable, to complete the patient-reported questionnaires.

Index event (t=0)

Anticoagulant prescribed
(t=+1 month)     t=+3 months     t=+6 months

t=-12 months

**Pre-index observation period (Group 1)**
- Demographics
- Clinical characteristics
- Medical history
- Clinical assessments

**Post-index observation period (Group 1)**
- First anticoagulation treatment prescribed post-stroke

**Post-DOAC observation period (Group 2)**
- DOAC prescribing
- Hospital resource use
- Patient reported outcomes (PRO's)
  - Baseline (anti-coagulant prescribed):
    - Bespoke questionnaire about experiences
  - 3 and 6 months post-anticoagulant prescription:
    - Morisky Medical Adherence Scale (MMAS-8)
    - Treatment Satisfaction Questionnaire for Medication (TQSM-9)
    - Bespoke adherence questionnaire

**Fig 1. Study eligibility, timeline and data collection.** Index event was defined as first stroke attributed to non-valvular atrial fibrillation. Key: DOAC – direct oral anticoagulant.

objective is evaluated only in Group 1. The secondary objectives of the study were only evaluated in Group 2 and were to describe the management pathways of patients, describe hospital resource use and clinical assessments associated with FXa-inhibiting DOAC treatment, and to describe the patient-reported satisfaction and experience of FXa-inhibiting DOAC treatment. Primary care data were requested from patients' primary care where relevant and were received from 64 General Practitioners (GPs) in total.

## Group 1 data collection

Data for the 12 months pre-index event and 1 month post-stroke were collected retrospectively from the hospital medical records of Group 1 patients to describe patient demographics (including age, sex, body mass index [BMI], ethnicity, smoking status, alcohol use), clinical characteristics (including time from AF diagnosis until stroke, recorded stroke severity, location of presentation with stroke, clinical and laboratory evaluations at the time of stroke (including liver function, renal function), relevant medical history (including all factors required to calculate the $CHA_2DS_2$-VASc risk score, renal disease, hepatic disease) and on the type of anticoagulant prescribed post-stroke [9].

## Group 2 data collection

For patients in Group 2, data on FXa-inhibiting DOAC treatment (including dose patterns and discontinuations) and hospital resource utilisation (including details of planned and unplanned hospital visits) related to AF or FXa-inhibiting DOACs were collected prospectively from hospital medical records for the six-month period following FXa-inhibiting DOAC initiation.

For patients in Group 2, patient-reported outcome data on adherence (Morisky Medication Adherence Scale [MMAS-8; for this study, question 8 was modified from the original wording such that it specified only about oral anticoagulation medications] [27–30], and a 4-question bespoke questionnaire to find out whether, how and how often patients had taken their DOAC medication in the previous 7 days), treatment satisfaction (Abbreviated Treatment Satisfaction Questionnaire for Medication [TSQM-9]) and general experience of treatment at the time of first dose of DOAC (a 6-question bespoke questionnaire to find out about communication, education about DOAC treatment and overall hospital experience) were collected via questionnaires [27,31].

The bespoke questionnaire regarding experience of treatment at the time of first FXa-inhibiting DOAC was administered within one month of the FXa-inhibiting DOAC being prescribed, while the MMAS-8, TSQM and bespoke adherence questionnaires were administered at 3- and 6-months after the initiation of FXa-inhibiting DOAC (with a + 1 month window to allow for completion; Fig 1). Questionnaires were administered face-to-face if the patient was attending clinic at the relevant time point, otherwise questionnaires were administered via telephone. Where the questionnaire was completed over the telephone, patients were posted the relevant questionnaires by NHS staff members prior to the telephone call to allow patients time to fill out the questionnaire and review their responses.

## Statistical analyses

As the study was planned to be descriptive, the study sample size was based upon precision estimates (95% confidence interval [CI]). No comparative analyses were performed. Precision estimates were performed based on: expected proportions of patients demonstrating certain characteristics, for example showing that with a sample of n = 250, of which 20% are shown to have vascular disease, the confidence limits are between 15%-25%, suggesting reliable estimates will be obtained with this sample size (Group 1; S6 Table in S1 File), for precision around the use of the MMAS-8 assuming a mean adherence score of 7.2 (based upon previously published work [32]), a sample of 40 patients was observed to give lower and upper confidence limits of 6.8 and 7.6 (standard deviation - 1.2) suggesting reliable estimates will be obtained with this sample size (Group 1; S7 Table in S1 File).

All analyses were descriptive in nature. Distributions and descriptive statistics of central tendency (medians and arithmetic or geometric means) and dispersion (SD, interquartile range [IQR], range) were presented for quantitative variables. Categorical variables were described with frequencies and percentages. Each analysis was only conducted from patients with available data in their medical records, with the total number of patients included specified alongside each result. Ordinal variables were evaluated using either frequencies and percentages or medians and IQRs or both, depending on the number of possible values for the variable. Body mass index (BMI) categories were calculated based upon numerical thresholds: underweight < 18.5, healthy weight: ≥ 18.5 to < 25, overweight ≥ 25 to < 30, obesity ≥ 30. The $CHA_2DS_2$-VASc score was calculated for each patient using the published scoring criteria [9]. MMAS-8 questionnaires were scored using the supplied algorithm and patients were classified as low adherence with a score of <6, medium adherence with a score of ≥6 and <8 and high adherence with a score of 8 [27]. Scores for the TSQM were derived following the published algorithm [31], whilst scores for the bespoke questionnaires on adherence and treatment experience were evaluated using frequencies and percentages.

## Results

### Patient demographics and clinical characteristics (Group 1)

234 patients were recruited from 8 centres (Group 1). Table 1 summarises the cohort baseline demographics for the overall cohort. Overall, 36% (n = 85/234) of patients were female. Mean (SD) age at index was 74.7 (10.1) years. Median

**Table 1. Demographic and clinical characteristics at time of first stroke (index) by anticoagulant prescribed.**

| Demographic and clinical characteristics | Overall |
|---|---|
| **N** | 234 |
| **Age, years (mean [±SD])** | 74.7 (10.1) |
| **Females (n, %)** | 85 (36%) |
| **Ethnicity (n, %)** | |
| Asian/Asian British | 4 (2%) |
| Black/Black British | 3 (1%) |
| Middle Eastern | 1 (0%) |
| Mixed | 2 (1%) |
| Pakistani | 1 (0%) |
| Poland | 1 (0%) |
| White | 210 (90%) |
| Not stated | 12 (5%) |
| **BMI categories (n, %)[*]** | |
| Healthy weight | 29 (12%) |
| Obesity | 52 (22%) |
| Overweight | 48 (21%) |
| Underweight | 2 (1%) |
| Not recorded | 103 (44%) |
| **Smoking status (n, %)** | |
| Current smoker | 23 (10%) |
| Ex-smoker | 91 (39%) |
| Non-smoker | 112 (48%) |
| Not recorded | 8 (3%) |
| **Alcohol use (n, %)[**]** | |
| Yes | 145 (62%) |
| No | 59 (25%) |
| Not recorded | 30 (13%) |
| **History of renal impairment (n, %)[**]** | |
| Yes | 25 (11%) |
| No | 206 (88%) |
| Not recorded | 3 (1%) |
| **Time from AF diagnosis to stroke (days, median [IQR; range])[***]** | 0.0 (-1.0-0.0; -485.0-4,575.0) |
| **$CHA_2DS_2$-VASc risk scores (2–9)[****]** | **n (%=224)** |
| 2 | 26 (12%) |
| 3 | 41 (18%) |
| 4 | 58 (26%) |
| 5 | 69 (31%) |
| 6 | 25 (11%) |
| 7 | 5 (2%) |

[*]Categories were based upon numerical thresholds: underweight < 18.5, healthy weight: ≥ 18.5 to < 25, overweight ≥ 25 to < 30, obesity ≥ 30

[**]Renal impairment and alcohol use (yes/no) was recorded as defined in medical records.

[***]Negative values indicate that the patient was diagnosed with AF after their stroke

Key: BMI – body mass index, AF – atrial fibrillation, IQR – inter-quartile range.

[****]In accordance with the scoring criteria, prior stroke adds +2 to the CHA2DS2-VASc score. Since all participants in this study had had a prior stroke (as a function of the inclusion criteria for the study) the minimum possible score at index was 2. CHA2DS2-VASc scores were only calculated for those patients for whom all components of the score were available (10 patients had missing data and thus did not have their score calculated).

(IQR) time from AF diagnosis until stroke was 0.0 (-1.0-0.0) days, where a negative number indicates that the diagnosis was made after the presentation of stroke. As recorded in the notes at index, 22% of patients were classified as obese (BMI ≥ 30, n = 52/234), 11% (n = 25/234) had history of renal impairment, 10% (n = 23/234) of patients were current smokers and 62% (n = 145/234) drank alcohol. $CHA_2DS_2$-VASc risk scores at index ranged from 2 to 7 (n = 224); 70% (157/224) had a score of >=4. It was not possible to calculate $CHA_2DS_2$-VASc risk score for 10 patients due to missing data for required components of the score (See S1 Table in S1 File for stratification by anticoagulant).

86% (n = 202/234) of patients presented with stroke at accident and emergency (A&E), 4% (n = 10/234) presented at general practitioners, 4% (n = 10/234) at an outpatient clinic, 2% (n = 4/234) presented at a hyper acute stroke unit and 3% (n = 8/234) at another setting. Stroke severity was documented by National Institutes of Health Stroke Scale (NIHSS; n = 176), 52% (92/176) of patients had a mild stroke, 34% (n = 60/176) were moderate, 11% (n = 20/176) were moderate to severe and 2% (n = 4/176) were severe. Of those with stroke classified by Oxfordshire Community Stroke Project score (n = 118), 58% (n = 69) patients were documented as having had partial anterior circulation infarcts (PACI), 22% (n = 26) as having lacunar infarcts (LACI), 12% (n = 14) as having posterior circulation infarcts (POCI) and 8% (n = 9) recorded total anterior circulation infarcts (TACI).

### Management pathway (Group 2)

Of the 93 patients recruited into Group 2, 49 were prescribed apixaban, 39 edoxaban, and 5 rivaroxaban. The median time from stroke onset to first dose of FXa-inhibiting DOAC was 6 (IQR, 2.0–10.2) days (n = 88). Of those patients prescribed apixaban, 87% (n = 40/46) of prescriptions were 5.0 mg twice daily, 11% (n = 5/46) were 2.5 mg twice daily, 2% (n = 1/46) were 15.0 mg twice daily; for edoxaban, 89% (n = 33/37) of prescriptions were 60.0 mg once a day, 11% (n = 4/37) were 30.0 mg once a day; for rivaroxaban, 80% (n = 4/5) of prescriptions were 20.0 mg once a day and 20% (n = 1/5) were 15.0 mg once a day. Of 92 patients, the choice of FXa-inhibiting DOAC was documented as having been informed by the physician's clinical opinion for 86% (n = 79) of patients. Other documented reasons included level of clinical experience with treatment (1%; n = 1), local guidelines (7%; n = 6) and patient choice (3%; n = 3). One patient was recorded as having had a dose change (increase) during the post-index observation period. Of the patients in Group 2 with available information in their medical records (n = 86/93), 99% (n = 85/86) were still being treated with their first FXa-inhibiting DOAC at the end of the study observation period; one patient was documented to have died at 3.8 months. No patients were initiated on nonsteroidal anti-inflammatory drugs (NSAIDs) or antiplatelets (including aspirin) during the post-index observation period.

### Hospital care resource utilisation (HCRU) and clinical assessments (Group 2)

Of the 93 patients in Group 2 with relevant available data in their medical records (n = 80/93), half (50%, n = 40/80; n = 13 had no data recorded) had at least one outpatient visit within the post-DOAC observation period related to AF or FXa-inhibiting DOACs, and of these, 40% (n = 16/40) had a single visit and 60% (n = 24/40) had 2 or more (Table 2). The mean (SD) time to first follow-up appointment from the date of first FXa-inhibiting DOAC initiation was 39.4 (42.3) days.

Of these 80 patients, 10% (n = 8) of patients had at least 1 A&E visit documented related to AF or FXa-inhibiting DOACs; 6% (n = 5) had a single A&E attendance whilst 4% (n = 3) had 2 attendances recorded during the post-index observation period. Ten reasons were given for attending A&E including AF, chest pain, post-stroke headache and a sprained ankle. Of 11 recorded A&E attendances, 36% (n = 4) resulted in an inpatient admission. Of 80 patients, 5 patients had at least one inpatient admission related to AF or FXa-inhibiting DOACs during the post-index observation period and one patient had 2 inpatient admissions. Of 5 inpatient admissions that were not ongoing at the time of data collection, the length of stay for 60% (n = 3/5) 20% (n = 1/5 and 20% (1/5) were 1 2 and 30 days, respectively.

**Table 2.** Hospital care resource utilisation (HCRU) - outpatient visits in the post-DOAC observation period.

| Hospital Resource Use – Outpatient visits | |
|---|---|
| **Proportion of patients with ≥1 outpatient clinic appointments during the post-DOAC observation period** | **n (%=80)** |
| Yes | 40 (50%) |
| None recorded | 40 (50%) |
| **Number of outpatient appointments during the post-DOAC observation period, per patient** | **n (%=40)** |
| 1 | 16 (40%) |
| 2 | 12 (30%) |
| 3 | 5 (12.5%) |
| 4 | 3 (7.5%) |
| ≥5 | 4 (10%) |
| **Distribution of setting for outpatient appointments** | **n (%=95)** |
| Face-to-face | 83 (87%) |
| Telephone | 11 (12%) |
| Virtual | 1 (1%) |
| **Reasons for outpatient appointments** | **n (%=110)** |
| Routine follow-up | 58 (53%) |
| Clinical assessment | 30 (27%) |
| Blood tests | 8 (7%) |
| Neurology | 5 (5%) |
| Other | 9 (8%) |

### Adherence (Group 2)

A total of 63 and 52 patients completed the MMAS-8 scale at 3- and 6-months post-index, respectively. MMAS-8 results are summarised in **Fig 2**.

68 patients filled out the bespoke adherence questionnaire at 3 months and 60 at 6 months post-index (S4 Table in S1 File). When asked if they'd missed any dose of their current anticoagulation medication over the past 7 days (Question 3), 3 patients (4%) of patients reported dose missing at 3 months post-initiation whereas none at the 6 month follow-up. Of the three patients reported missing dose, two were due to the patient forgetting and one due to feeling unwell (reporting nausea, tiredness and fatigue).

### Patient-reported satisfaction with treatment and care (Group 2)

All patients completing question 9 of the TSQM at 3 (n=64) and 6 months (n=51) reported being either somewhat satisfied (3 months – 17%, n=11; 6 months – 14%, n=7), satisfied (3 months – 23%, n=15; 6 months – 29%, n=15), very satisfied (3 months – 39%, n=25; 6 months – 43%, n=22) or extremely satisfied (3 months – 20%, n=13; 6 months – 14%, n=7) with their anticoagulant medication. No patients reported being dissatisfied at 3- or 6-month post-DOAC initiation.

Of 57 patients who completed the baseline care satisfaction questionnaire, 65% (n=37) of patients reported that they were extremely satisfied with their care and 35% (n=20) reported satisfied (See S5 Table in S1 File for a summary of questionnaire responses). When asked how confident they were that they know how to take their treatment, 68% (n=39) reported very confident, 28% (n=16) were fairly confident and 4% (n=2) reported that they were neither confident nor not confident. When asked how confident they were that they know how often to take their treatment, 74% (n=42) of patients reported that they were very confident, 23% (n=13) were fairly confident and 4% (n=2) were neither confident nor not confident.

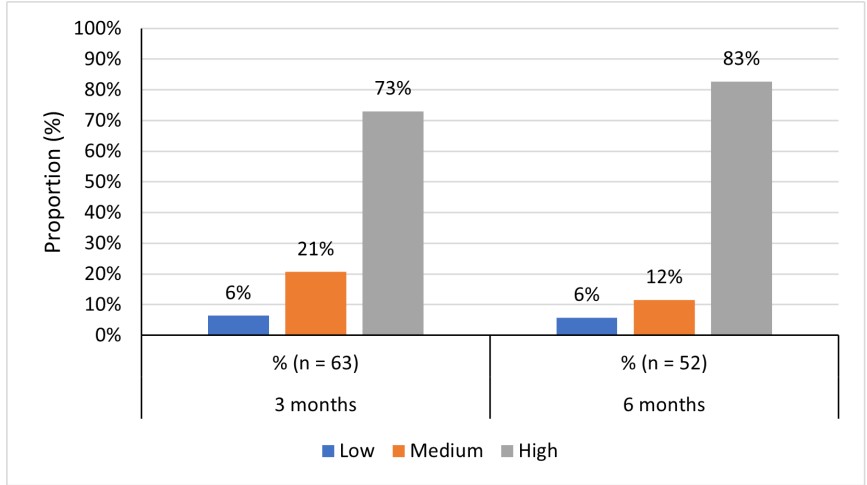

**Fig 2. Adherence as measured by the MMAS-8 scale at 3 and 6 months post-DOAC initiation.**

## Discussion

This study provides insight into the characteristics of UK patients first initiated on DOAC treatment following an ischaemic stroke, and found high levels of adherence, persistence, and treatment satisfaction during the 6 months post-DOAC initiation. The observed baseline demographics and clinical characteristics (including $CHA_2DS_2$-VASc scores) were broadly in line with published real-world studies of DOAC use in patients with AF [24,33–35]. The observed time from stroke to initiation of FXa-inhibiting DOAC varied widely, with the majority initiated within the acute phase. This may be due to differences in stroke characteristics (e.g., variation in stroke severity) or variation in clinical practice. Reported numbers of outpatient attendances are low versus those expected in accordance with national guidelines [36], although these may have been affected by the COVID-19 pandemic. Most patients were prescribed FXa-inhibiting DOAC at the suggested dosages (as per the respective guidelines), with very low rates of dose changes or interruptions over the post-index observation period, suggesting strong medication adherence, in line with the patient-reported measures of adherence (MMAS-8).

The overall distribution of baseline demographics and clinical characteristics (including $CHA_2DS_2$-VASc scores) in this study appear broadly in line with that seen in several real-world studies of DOAC use in patients with NVAF [24,33–35]. The cohorts in the cited studies included patients both with and without prior stroke, whilst patients in this study had to have had a stroke to be eligible. Notably, more than two thirds of patients had a $CHA_2DS_2$-VASc score of 4 or more. Eighty-six percent of patients presented with stroke at A&E having never been diagnosed with AF, this finding suggests that a significant proportion of patients who had a stroke attributable to NVAF would have potentially been eligible for anticoagulation for primary prevention had AF been diagnosed prior to the stroke, supporting the case for AF screening initiatives in groups at high risk of AF (30). Further research that describes hospital care resource utilisation (HCRU) in the months prior to stroke may help understand if there are missed opportunities to identify and treat AF earlier.

Whilst the median time between stroke and FXa-inhibiting DOAC initiation was 6 days, it ranged from 0 days (indicating the FXa-inhibiting DOAC was initiated on the day of the stroke) to 485 days post-stroke. The reason for the significant delay to FXa-inhibiting DOAC initiation is not known and may relate to later increase in embolic risk and/or fall in bleeding risk. It is also possible that the larger values observed here relate to the use of subsequent Holter monitoring, although this was not specifically investigated within this study and publications evidence the importance of early investigations for AF and its subsequent impact on early anticoagulant prescriptions [37,38]. Guidance from the Royal College of Physicians recommends that anticoagulants are initiated within 2 weeks of ischaemic stroke for patients with known AF and

no haemorrhagic transformation [39]. Observational data suggest that early anticoagulation initiation post-stroke may be beneficial in selected patients [40,41], however primary evidence to inform the appropriate time of initiation of anticoagulation post-stroke is lacking. Recent results from the ELAN trial suggest that early DOAC initiation (<48 hours after minor or moderate stroke, and on day 6 or 7 following major stroke) may be appropriate in acute stroke patients with NVAF [42]. Ongoing studies such as the OPTIMAS [43] are further exploring this issue.

The most common reason given for FXa-inhibiting DOAC choice was clinical opinion (86%), whilst patients' choice (3%) and experience with treatment (1%) were much less common. Given that the guidelines for all three FXa-inhibiting DOAC recommend the patient be involved in the decision making [10,12,13], and that there is apparent parity between the FXa-inhibiting DOACs there may be a need for more research into the reasons why so few clinicians recorded their choice of FXa-inhibiting DOAC due to experience with treatment, although this could have been influenced by local guidelines. This observed results also suggest that in the absence of head-to-head trials of DOACs to guide clinical judgement, there may be a need for more clinician education on FXa-inhibiting DOAC to overcome a perceived lack of experience (in addition to wider factors such as the general health and compliance of patients receiving treatment).

The majority (between 80%-89%) of patients observed received the recommended doses of apixaban, edoxaban and rivaroxaban as per UK guidelines [12,13]. The proportion of patients observed to have received a reduced FXa-inhibiting DOAC dosage (11%-20%) is lower than previously published real-world DOAC use, for example, the ETNA-AF-EUROPE study reported 23.4% of patients receiving the reduced dose of edoxaban, whilst in the XANTUS study, 20.8% of patients receiving rivaroxaban were reported to have received the reduced dosage [24,33]. Only 1 patient was documented to have had a change in dose and 1 patient had discontinued their first treatment by 6 months (due to death), thus evidencing high levels of treatment persistence with the prescribed FXa-inhibiting DOAC. A previous UK population–level study also observed high levels of persistence to FXa-inhibiting DOAC, reporting proportions of 83.1% and 81.8% at 6 months and 70.3% and 68.1% at 12 months for those taking apixaban and rivaroxaban, respectively [44]. One real-world study of rivaroxaban use reported 8.8% of patients to have experienced at least 1 interruption to therapy and 80% of patients to have persisted on therapy up to a year post initiation [24]. This study had a shorter follow-up of 6 months, and as such the results are not directly comparable. Given that the COVID-19 pandemic overlapped with this study, there may have been an impact on patients' ability to review their medication with their consulting clinician.

Regarding HCRU, only 50% of patients had 1 or more outpatient clinic appointments recorded in the post-DOAC observation period. DOACs require less frequent follow up compared with warfarin which requires frequent follow up for INR checks and thus, the use of DOACs may be associated with reduced medical costs as has been suggested in published studies comparing the cost effectiveness of DOACs vs warfarin [4,45,46]. Again, the COVID-19 pandemic may have impacted on follow up reviews.

Adherence to FXa-inhibiting DOAC as measured by the MMAS-8 was broadly consistent across the 3- and 6-month time points post-initiation, with the only observable change being that there was an approximate 10% increase in patients with high adherence from 3 to 6 months. This may suggest that adherence improved over time as patients became more familiar with the routine of taking their medication, with adherence peaking at 83% at month 6, but may have been influenced by the study contact/clinic visits at months 3 and 6 where the questionnaire was recorded. The adherence rates observed in this study differ slightly when compared to other studies conducted in other European countries, though this likely reflects differences in the study cohorts and methods used to measure adherence [34,44,47].

In this study, patient-reported satisfaction with FXa-inhibiting DOAC treatment was positive, with no patients reporting being dissatisfied at either the 3- or 6-month time point post-DOAC initiation. This is in line with positive findings of independent studies of treatment satisfaction that also found that DOACs were either found to be similarly satisfactory to treatment with warfarin [48], or more satisfactory than warfarin, although the observed levels of treatment satisfaction were not reported to impact on adherence [49].

## Limitations

The sample size of the study was smaller than planned; recruitment was interrupted by the COVID-19 pandemic and study recruitment was halted before the target sample sizes were achieved for Group 1 and the rivaroxaban subgroup (Group 2) as forecasts showed it would take too long to achieve the required sample size. The sample sizes for Group 1 and the apixaban/edoxaban subgroups (Group 2) are large enough to provide reliable mean estimates, however the rivaroxaban subgroup is too small for any meaningful inferences to be made. Due to the overlap in time between the study and the COVID-19 pandemic, it should be noted that there may have also been an impact on the observed resource utilisation and subsequent patient follow-up. Due to some components of the $CHA_2DS_2$-VASc score not being available for 10 patients, some bias may have been introduced due to a reduced sample size or by underestimation of stroke risk. As it was not a predefined outcome, demographics were not specifically provided for those patients in Group 1 who were diagnosed with NVAF but had not received anticoagulation therapy prior to first stroke, however future research should attend to these patients. We were unable to link the initially prescribed doses of FXa-inhibiting DOACs to creatinine clearance data at baseline due to a low number of records being available which may introduce bias due to small sample size (<15 records available). Data from primary care once patients had left secondary care or relating to management prior to the first stroke (index event) were not available for all patients (64 GPs provided data in total) and may have affected related endpoints. The study was descriptive and no analyses to control for confounding factors were carried out. Subsequently, no inferences can be made regarding the relationships between DOACs and study endpoints. Real-world data from medical records and patient-reported measures rely on the completeness of medical records and/or the given answers; these were not queried or followed up to clarify inconsistencies or complete missing answers. Whilst the importance of providing complete and honest answers was stressed to participants, the data may be subject to reporting bias. Additionally, patient-reported data such as the adherence to medication questionnaire is based on recall and may therefore be subject to recall and/or reporting bias.

## Conclusions

This study provides real-world evidence of the patient characteristics, management pathways and patient-reported experiences of patients prescribed an FXa-inhibiting DOAC (apixaban, edoxaban, rivaroxaban) after first stroke attributable to NVAF in UK clinical practice. DOAC usage was largely in compliance with guidelines, and significant variation in time to initiation post stroke was noted. High levels of adherence, persistence, and treatment satisfaction in the 6 months post-initiation of FXa-inhibiting DOAC were demonstrated. There is potential for screening programs to identify AF prior to first stroke and initiate stroke prevention treatment with anticoagulation.

These results provide clinicians with valuable insights into the experience of post-stroke patients with NVAF receiving FXa-inhibiting DOACs treatment for secondary prevention of stroke during the 6 months post-stroke.

## Supporting information

**S1 File. Supplementary materials.** This document contains supplementary tables 1–7.
(DOCX)

## Disclaimer

The MMAS-8 Scale, content, name, and trademarks are protected by US copyright and trademark laws. Permission for use of the scale and its coding is required. A license agreement is available from MMAR, LLC., www.moriskyscale.com.

Author note: NB. Morisky,D.E., Ang,A., Krousel-Wood, M. and Ward,H.J.(2008), Predictive Validity of a Medication Adherence Measure in an Outpatient Setting. The Journal of Clinical Hypertension,10:348–354.https://doi.org/10.1111/j.1751-7176.2008.07572.x. This article has been retracted by agreement between the journal's Editor-in-Chief, Dr.

Ji-Guang Wang, and Wiley Periodicals LLC. Following publication, concerns were raised by a third party regarding the statistical analysis presented in the article. Following an independent statistical review the Journal no longer has confidence in the reported conclusions and issued a retraction 08/2023.

## Author contributions

**Conceptualization:** James Uprichard, Chris Hatton, Amelia Reed.

**Formal analysis:** Joe Eva.

**Investigation:** James Uprichard, Liqun Zhang, Anand Dixit, Yaqoob Bhat, Amit Mistri, Dipankar Dutta, Khalid Rashed, Dumin Karunatilake.

**Methodology:** James Uprichard, Chris Hatton, Joe Eva, Amelia Reed.

**Project administration:** James Uprichard, Chris Hatton, Amelia Reed.

**Resources:** Chris Hatton, Joe Eva, Amelia Reed.

**Supervision:** James Uprichard, Chris Hatton, Amelia Reed.

**Visualization:** Joe Eva.

**Writing – original draft:** James Uprichard, Liqun Zhang, Anand Dixit, Yaqoob Bhat, Amit Mistri, Dipankar Dutta, Khalid Rashed, Dumin Karunatilake, Chris Hatton, Joe Eva, Amelia Reed.

**Writing – review & editing:** James Uprichard, Liqun Zhang, Anand Dixit, Yaqoob Bhat, Amit Mistri, Dipankar Dutta, Khalid Rashed, Dumin Karunatilake, Chris Hatton, Joe Eva, Amelia Reed.

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
