## [Decision Letter · Decision Letter 0]

4 Nov 2024

PONE-D-24-30678Real-world management, resource use, patient-reported outcomes and adherence in patients treated with direct oral anticoagulants for a first stroke attributable to non-valvular atrial fibrillation: a UK mixed-methods non-interventional studyPLOS ONE

Dear Dr. Uprichard,

Thank you for submitting your manuscript to PLOS ONE. After careful consideration, we feel that it has merit but does not fully meet PLOS ONE’s publication criteria as it currently stands. Therefore, we invite you to submit a revised version of the manuscript that addresses the points raised during the review process.

**ACADEMIC EDITOR: ** Thank you for submitting your manuscript to the Plos One. After careful consideration, this manuscript has merit but does not fully meet journal publication criteria as it currently stands. Therefore, we invite you to submit a revised version of the manuscript addressing the concerns the reviewers raised, specifically regarding study methodology and the clarity of your presentation. Please see the attached reviewer comments and details below.

We look forward to receiving your revised manuscript.

Kind regards,

Dr Redoy Ranjan, MBBS, MRCSEd, Ch.M., MS (CV&TS), FACS

Academic Editor

PLOS ONE

Journal Requirements:

[JU reports having received support for the present manuscript including sponsorship of the study and medical writing support from Daiichi Sankyo. AM reports having received payment or honoraria from Daiichi Sankyo; support for attending meetings and/or travel from Daiichi Sankyo. CH and AR are employees of Daiichi Sankyo UK. JE was an employee of OPEN Health, London for the duration of the study. KR, LZ, AD, YB, DD, and DK report no conflicts of interest to declare.].

Reviewers' comments:

Reviewer's Responses to Questions

**Comments to the Author**

1. Is the manuscript technically sound, and do the data support the conclusions?

Reviewer #1: Yes

Reviewer #2: Partly

Reviewer #3: Partly

2. Has the statistical analysis been performed appropriately and rigorously? 

Reviewer #1: Yes

Reviewer #2: I Don't Know

Reviewer #3: No

3. Have the authors made all data underlying the findings in their manuscript fully available?

Reviewer #1: Yes

Reviewer #2: Yes

Reviewer #3: Yes

4. Is the manuscript presented in an intelligible fashion and written in standard English?

Reviewer #1: Yes

Reviewer #2: Yes

Reviewer #3: Yes

5. Review Comments to the Author

Reviewer #1: Well presented manuscript regarding the use of anticoagulants in patients with stroke presenting real world data.

Adherence to the guidelines is important and it is highlighted on the current paper. Good quality of data.

Reviewer #2: This is a well-written manuscript that describes a mixed-methods study investigating the use of direct oral anticoagulants (DOACs) in UK patients with a first stroke attributable to non-valvular atrial fibrillation (NVAF). The study provides valuable insights into patient characteristics, treatment patterns, adherence, and patient-reported outcomes.

Strengths:

The study clearly addresses the need for real-world data on DOAC use in stroke patients with NVAF. The study recruited a sufficient number of patients to draw meaningful conclusions which serves the owner of the study.

Weaknesses:

The study only followed patients for 6 months, which may not be sufficient to capture long-term adherence and outcomes.

It does not explore the reasons for the wide variation in time to DOAC initiation.: Not all patients completed all questionnaires, which could introduce bias.

Recommendations:

Abstract: The abstract could be strengthened by highlighting the key findings regarding adherence, satisfaction, and resource utilization.

Introduction: The authors could expand on the potential benefits and risks of DOAC use in this patient population.

Methods: The authors should clarify how missing data was handled in the analyses.

Results: The results section could be more concise by focusing on the most important findings. Consider presenting some data in tables or figures for better visualization.

Discussion: The discussion should elaborate on the potential reasons for delayed DOAC initiation and the implications for clinical practice.

Limitations: The limitations section should acknowledge the potential impact of the COVID-19 pandemic on resource utilization and patient follow-up.

Overall Recommendation:

This is a good quality study that deserves to be published in a peer-reviewed journal with minor revisions.

Reviewer #3: The authors present descriptive statistics of characteristics and outcomes of adult patients with nonvalvular atrial fibrillation (NVAF) receiving direct oral anticoagulants (DOACs) for secondary stroke prevention at secondary care center in United Kingdom (UK). Although this is interesting data, there are several issues which need to be clarified.

<comments>

1. The title should include the fact that this study is based on real-world data from secondary care centers in UK.

2. As the author points out, it is important to identify AF before a stroke occurs. Considering that the minimum CHA2DS2vasc score is 2 points in group 1, the fact that anticoagulant therapy was not provided despite having the diagnosis of NVAF is a serious situation. The authors should describe the proportion of such patients and their background why they were not provided anticoagulation therapy despite having NVAF. It would also be desirable to show the proportion of patients who had already been receiving anticoagulant treatment for some reason but then developed a stroke.

3. Of the 93 registered patients in group 2, only 60 (64.5%) patients completed the 6-month bespoke adherence questionnaire. The authors should indicate the follow-up rate and DOACs prescription rate at 6 months. Since it is highly likely that DOACs are discontinued in patients who drop out, the authors should state the possibility of extremely low adherence in such cases.

4. The author should clearly describe the method of obtaining consent in Methods section of the manuscript. In particular, the author should clearly state whether consent was obtained from all participants, including Group 1.

5. The authors should clearly state the inclusion and exclusion criteria for this study in Methods section of the manuscript.

6. Because the patients who have stroke occurring despite anticoagulant treatment being provided are considered clinically important, the author should clearly state the rational for excluding them.

7. The absence of patients who is treated by pulmonary vein isolation (PVI) is unnatural in recent era. If the PVI is to be excluded, the author should clearly state PVI as an exclusion criterion. In that case, the author should clearly state the background why PVI is not indicated.

8. The authors should clearly state the rational reason for excluding dabigatran in group 2. In addition, the author should add the demographics of group 2.

9. The creatinine clearance (Ccr) value is very important when prescribing DOACs. Therefore, the creatinine clearance value at the beginning of DOACs and at the end of the observation period should be added. Similarly, the author is encouraged to add the percentage of patients who were prescribed DOACs at inappropriate doses.

10. The author should state the basis for the sample size setting in Statistical analyses of Methods section. In addition, the author should also state that this study did not reach

11. In Table 1, the 10 cases that could not be calculated CHA2DS2casc score should be included in the table 1.

12. The definition of body mass index (BMI) categories should be described in Methods section.

13. The definition of renal impairment should be clearly stated in Methods section. In particular, it should be clearly stated whether the definition includes urinary protein or is based solely on eGFR. In addition, since the DOAC dosage is set based on Ccr, Ccr should also be stated.

14. The definition of alcohol consumption should be clearly stated in Methods section. Some papers have suggested a link between heavy drinking and AF. It should be clearly stated whether this includes only heavy drinkers or also occasional drinkers who drink less than twice a week.

15. The author should add the study flow diagram of this study. In particular, since the population differs for hospital care resource utilisation, MMAS-8 scale, and TSQM-9, it is desirable to include the population parameters for the available data in the diagram.

16. It is stated that 86 people completed the scheduled observation period for group 2. The details of the 7 people who dropped out should be clearly stated in the Result section and study flow diagram.

17. The use of unnecessary and uncommon abbreviations should be discouraged.</comments>

6. PLOS authors have the option to publish the peer review history of their article (what does this mean? ). If published, this will include your full peer review and any attached files.

**Do you want your identity to be public for this peer review?** For information about this choice, including consent withdrawal, please see our Privacy Policy .

Reviewer #1: **Yes: ** Afendoulis Dimitrios

Reviewer #2: **Yes: ** Maha Ahmed

Reviewer #3: No

---

## [Author Response · Author response to Decision Letter 1]

22 Dec 2024

Response to reviewer comments:

We thank the reviewers for kindly spending the time to review this manuscript and providing their input. Below we have addressed the reviewers’ comments (Italicized below) and numbered them. Within the ‘marked’ copy of the revised manuscript, these changes can be found as tracked changes, in addition to being highlighted in yellow and with associated comments that contain the relevant comment number within this document.

Reviewer 1:

Well presented manuscript regarding the use of anticoagulants in patients with stroke presenting real world data.

Adherence to the guidelines is important and it is highlighted on the current paper. Good quality of data.

We thank the reviewer for their positive comments.

Reviewer 2:

This is a well-written manuscript that describes a mixed-methods study investigating the use of direct oral anticoagulants (DOACs) in UK patients with a first stroke attributable to non-valvular atrial fibrillation (NVAF). The study provides valuable insights into patient characteristics, treatment patterns, adherence, and patient-reported outcomes.

Strengths:

The study clearly addresses the need for real-world data on DOAC use in stroke patients with NVAF. The study recruited a sufficient number of patients to draw meaningful conclusions which serves the owner of the study.

Weaknesses:

The study only followed patients for 6 months, which may not be sufficient to capture long-term adherence and outcomes.

It does not explore the reasons for the wide variation in time to DOAC initiation.: Not all patients completed all questionnaires, which could introduce bias.

Recommendations:

1. Abstract: The abstract could be strengthened by highlighting the key findings regarding adherence, satisfaction, and resource utilization.

The key findings around adherence and satisfaction are already highlighted within the abstract. Specifically, that “At 3 and 6 months, 73% (46/63) and 83% (43/52) had high (score of 8) Morisky Medication Adherence Scale score (MMAS-8), respectively. No patients reported being dissatisfied at 3 or 6 months post-DOAC initiation.”

Regarding the addition of resource utilization results, we have added the following text (highlighted yellow below and in the tracked manuscript):

“For Group 2, the median time from stroke to first DOAC dose was 6 (IQR, 2.0–10.2; n=88) days; 50% patients had ≥1 outpatient visit recorded related to AF or DOACs.”

2. Introduction: The authors could expand on the potential benefits and risks of DOAC use in this patient population.

Text has been added to the introduction to better outline the potential benefits and risks of DOAC use in the studied patients as below:

“The increasing use of DOACs over vitamin K antagonists is likely reflective of their relative benefits including an improved efficacy:safety ratio, a favourable bleeding profile (although the risk of gastrointestinal bleeding is higher than that observed for warfarin 1), a reduced need for routine monitoring and fewer drug interactions 2.”

3. Methods: The authors should clarify how missing data was handled in the analyses.

It is specified in the section “Statistical analyses” that “Each analysis was only conducted from patients with available data in their medical records, with the total number of patients included specified alongside each result.” Missing data is common in real-world studies as there is no way to mandate all data to be collected in clinic, thus we accurately report the descriptive statistics for the data that has been collected and provide the relevant number of patients that it has been collected from.

4. Results: The results section could be more concise by focusing on the most important findings. Consider presenting some data in tables or figures for better visualization.

We thank the reviewer for their suggestions. However, we believe that the current draft of the results is appropriately concise given the granularity of the real-world data and that the current number of figures (1 results figure) and tables (2 tables) is pertinent to highlight those pieces of key information.

5. Discussion: The discussion should elaborate on the potential reasons for delayed DOAC initiation and the implications for clinical practice.

We thank the reviewer for this suggestion. There is already a paragraph on this in the discussion that outlines the observed levels of delay between stroke and DOAC initiation and reasons for delay/clinical implications. The paragraph is copied below for ease of review (and commented on in the manuscript).

“Whilst the median time between stroke and DOAC initiation was 6 days, it ranged from 0 days (indicating the DOAC was initiated on the day of the stroke) to 485 days post-stroke. The reason for the significant delay to DOAC initiation is not known and may relate to later increase in embolic risk and/or fall in bleeding risk. It is also possible that the larger values observed here relate to the use of subsequent Holter monitoring, although this was not specifically investigated within this study and publications evidence the importance of early investigations for AF and its subsequent impact on early anticoagulant prescriptions [35,36]. Guidance from the Royal College of Physicians recommends that anticoagulants are initiated within 2 weeks of ischaemic stroke for patients with known AF and no haemorrhagic transformation [37]. Observational data suggest that early anticoagulation initiation post-stroke may be beneficial in selected patients [38,39], however primary evidence to inform the appropriate time of initiation of anticoagulation post-stroke is lacking. Recent results from the ELAN trial suggest that early DOAC initiation (<48 hours after minor or moderate stroke, and on day 6 or 7 following major stroke) may be appropriate in acute stroke patients with NVAF [40]. Ongoing studies such as the OPTIMAS [41] are further exploring this issue. “

6. Limitations: The limitations section should acknowledge the potential impact of the COVID-19 pandemic on resource utilization and patient follow-up.

We thank the review for their suggestion. We have added the following statement to the limitations section in the discussion:

“Due to the overlap in time between the study and the COVID-19 pandemic, it should be noted that there may have also been an impact on the observed resource utilisation and subsequent patient follow-up.”

Overall Recommendation:

This is a good quality study that deserves to be published in a peer-reviewed journal with minor revisions.

We thank reviewer 2 for their comments and suggestions.

Reviewer 3:

The authors present descriptive statistics of characteristics and outcomes of adult patients with nonvalvular atrial fibrillation (NVAF) receiving direct oral anticoagulants (DOACs) for secondary stroke prevention at secondary care center in United Kingdom (UK). Although this is interesting data, there are several issues which need to be clarified.

7. The title should include the fact that this study is based on real-world data from secondary care centers in UK.

The title already contained the wording “real-world” but we have also now added in wording around the secondary care centres as highlighted below:

“Real-world management, resource use, patient-reported outcomes and adherence in patients receiving direct oral anticoagulants for first stroke attributed to non-valvular atrial fibrillation in secondary care: a UK mixed-methods observational study“

8. As the author points out, it is important to identify AF before a stroke occurs. Considering that the minimum CHA2DS2vasc score is 2 points in group 1, the fact that anticoagulant therapy was not provided despite having the diagnosis of NVAF is a serious situation. The authors should describe the proportion of such patients and their background why they were not provided anticoagulation therapy despite having NVAF. It would also be desirable to show the proportion of patients who had already been receiving anticoagulant treatment for some reason but then developed a stroke.

We thank the reviewer for their very important point.

Regarding the proportion of patients who had not been provided anticoagulants prior to stroke but had already received a diagnosis of non-valvular AF, we note that this is information of interest but that this was not an intended outcome of this study (please see the published study protocol)3. However, in order to provide additional information beyond the median (IQR) time from diagnosis of AF, we have additionally provided the range of days between diagnosis and stroke to evidence how widely this ranged (see below excerpt from Table 1 highlighted):

Time from AF diagnosis to stroke (days, median [IQR; range])** 0.0 (-1.0-0.0; -485.0-4,575.0)

Regarding the proportion of patients who had already been receiving anticoagulant treatment, the study was designed with inclusion criteria to specifically recruit patients who had not received anticoagulants in the 12 months prior to their stroke and thus this data is not available from this dataset.

9. Of the 93 registered patients in group 2, only 60 (64.5%) patients completed the 6-month bespoke adherence questionnaire. The authors should indicate the follow-up rate and DOACs prescription rate at 6 months. Since it is highly likely that DOACs are discontinued in patients who drop out, the authors should state the possibility of extremely low adherence in such cases.

Regarding the follow-up rate and DOACs prescription rate at 6 months, this information is outlined in the section “Management pathway (Group 2)”. We state that of the 93 subjects in group 2, 86/93 patients had available information in their medical records about their DOAC treatment to be able to say whether they were or weren’t on treatment still at the end of the 6 months observation period. Of the 86 patients with available information, 85/86 (99%) were still being treated with their first DOAC at the end of the 6 months observation period; one patient was documented to have died within the 6 month observation period. This information is highlighted and commented for reference in the marked up manuscript.

Regarding the completion rate of the adherence questionnaires, it is common in real-world observational studies for subject numbers to drop-off over time. Although it should be noted firstly that at the first time point of 3 months, 68 patients complete the questionnaire suggesting that of those that completed the questionnaire initially, a high percentage (60/68, 88%) return the questionnaire at the latter time point.

Lastly, given that the recorded levels of DOAC discontinuation (within the 6 month observation period) are very low for those patients whom had data available, the numbers responding to the questionnaire are not indicative or related to the numbers of patients maintaining treatment and thus we suggest that there is no need to state the possibility of low adherence in these cases.

10. The author should clearly describe the method of obtaining consent in Methods section of the manuscript. In particular, the author should clearly state whether consent was obtained from all participants, including Group 1.

We have added the following highlighted text to the methods section in order to provide further clarity over consent:

“Patients were contacted and recruited by members of their direct care team between 23 January 2019 and 30 June 2021; recruitment was paused between April 2020 to June 2020 due to the COVID-19 pandemic. All patients gave informed consent to take part in the study.”

11. The authors should clearly state the inclusion and exclusion criteria for this study in Methods section of the manuscript.

We thank the reviewer for raising this important point. The inclusion/exclusion criteria are laid out in the protocol manuscript but have now been reiterated in this publication text. The following text has been added into the methods section as highlighted below:

“All patients were eligible for inclusion in the study if they presented to the study centre with a first ischaemic stroke which was, in the clinician’s opinion, attributable to NVAF and were aged ≥18 years at the time of first stroke. In addition, to have been eligible for Group 2, patients in Group 1 must have been initiated on apixaban, edoxaban or rivaroxaban after their first stroke. All patients were excluded from participation in the study if they were unwilling or unable to give written informed consent, if their medical records were not available for review, if they had been prescribed anticoagulants for any purpose in the 12 months prior to the stroke diagnosis, if they had prior or current haemorrhagic stroke, if they had a diagnosis of transient ischaemic attack, and/or if they had a severe cognitive or emotional deficit. Patients from Group 1 were excluded from Group 2 if they were unwilling or unable to complete the patient-reported questionnaires.”

12. Because the patients who have stroke occurring despite anticoagulant treatment being provided are considered clinically important, the author should clearly state the rational for excluding them.

This criteria was agreed upon at the period of study design as there was emerging real-world data around the use of DOACs in patients with AF, however there was a clear knowledge gap at the time around the use of DOACs in patients at the time of first ischaemic stroke.3 Thus, whilst we acknowledge the clinical importance of these patients, the study was designed to fill a different gap in clinical knowledge.

13. The absence of patients who is treated by pulmonary vein isolation (PVI) is unnatural in recent era. If the PVI is to be excluded, the author should clearly state PVI as an exclusion criterion. In that case, the author should clearly state the background why PVI is not indicated.

We thank the reviewer for raising this important point. No ablation was recorded as treatment in the patients included in this study, this is likely because the majority of patients had their AF diagnosed on (or around) the date of their index stroke and subsequently, as per current NICE guidance in the UK, were started on anticoagulants in the first instance for secondary stroke prevention.4 NICE guidance currently states the use of catheter ablation (as PVI is more commonly termed in the UK) is an option if “drug treatment is unsuccessful, unsuitable or not tolerated” meaning that it is likely that any patients that required ablation, may have received this treatment after the study observation period as the majority of patients with data available (99%, n=85/86) remained on their first prescribed DOAC treatment at the end of the study observation period.4

Ablation was not part of the study exclusion criteria, nor was it specifically not recorded. We strongly feel that this subsequently does not warrant additional text in the manuscript introductory section as it is not the focus of the study and manuscript.

14. The authors should clearly state the rational reason for excluding dabigatran in group 2. In addition, the author should add the demographics of group 2.

The study focused on the use of direct anti-FXa inhibitors whilst dabigatran is a thrombin (factor IIa) inhibitor as written in the concluding paragraph of the introduction. Thus, dabigatran was not included in Group 2.

As per the study design laid out in the published study protocol,3 the demographics for Group 2 are not provided separately to Group 1.

15. The creatinine clearance (Ccr) value is very important when prescribing DOACs. Therefore, the creatinine clearance value at the beginning of DOACs and at the end of the observation period should be added. Similarly, the author is encouraged to add the percentage of patients who were prescribed DOACs at inappropriate doses.

We agree with the reviewer that CCr is an important measure when prescribing DOACs. Unfortunately, only a very small number of patients had baseline information available in their medical records (14 of 234) and thus it was decided not to include this information in the manuscript as it could be heavily biased due to its small sample size. Additionally, due to the way the data was recorded for the post-DOAC observation period, only 4 patients’ data is available and i

---

## [Decision Letter · Decision Letter 1]

12 Jan 2025

PONE-D-24-30678R1Real-world management, resource use, patient-reported outcomes and adherence in patients receiving direct oral anticoagulants for first stroke attributed to non-valvular atrial fibrillation in secondary care: a UK mixed-methods observational studyPLOS ONE

Dear Dr. Uprichard,

Thank you for submitting your manuscript to PLOS ONE. After careful consideration, we feel that it has merit but does not fully meet PLOS ONE’s publication criteria as it currently stands. Therefore, we invite you to submit a revised version of the manuscript that addresses the points raised during the review process.

**ACADEMIC EDITOR: **

After a critical external peer review by two experts, I recommended a minor revision to improve the paper's clarity and presentation based on the reviewers' concerns. Please see the attached reviewer comments below.

We look forward to receiving your revised manuscript.

Kind regards,

Dr Redoy Ranjan, MBBS, MRCSEd, Ch.M., MS (CV&TS), FACS

Academic Editor

PLOS ONE

Journal Requirements:

Reviewers' comments:

Reviewer's Responses to Questions

**Comments to the Author**

1. If the authors have adequately addressed your comments raised in a previous round of review and you feel that this manuscript is now acceptable for publication, you may indicate that here to bypass the “Comments to the Author” section, enter your conflict of interest statement in the “Confidential to Editor” section, and submit your "Accept" recommendation.

Reviewer #1: All comments have been addressed

Reviewer #2: All comments have been addressed

Reviewer #3: All comments have been addressed

2. Is the manuscript technically sound, and do the data support the conclusions?

Reviewer #1: Yes

Reviewer #2: Yes

Reviewer #3: Yes

3. Has the statistical analysis been performed appropriately and rigorously? 

Reviewer #1: Yes

Reviewer #2: I Don't Know

Reviewer #3: Yes

4. Have the authors made all data underlying the findings in their manuscript fully available?

Reviewer #1: Yes

Reviewer #2: Yes

Reviewer #3: Yes

5. Is the manuscript presented in an intelligible fashion and written in standard English?

Reviewer #1: Yes

Reviewer #2: Yes

Reviewer #3: Yes

6. Review Comments to the Author

Reviewer #1: Very well presented manuscript.

Given the inclusion of the review comments it is suitable for publication

Reviewer #2: This study investigates the use of direct oral anticoagulants (DOACs) in adult patients with non-valvular atrial fibrillation (NVAF) who had a first ischaemic stroke. The study included 234 patients in Group 1 and 93 in Group 2. Group 2 patients were initiated on apixaban, edoxaban, or rivaroxaban post-first stroke. The primary objective was to describe patients' demographics, clinical characteristics, and medical history, stratified by the anticoagulant prescribed. The secondary objectives were to describe patient management pathways, hospital resource use, clinical assessments associated with DOAC treatment, and patient-reported satisfaction and experience of DOAC treatment.

The results demonstrate high levels of adherence, persistence, and treatment satisfaction in the 6 months post-initiation of DOAC after first stroke attributable to NVAF. Patients were largely initiated on DOACs in compliance with guidelines, with most patients prescribed DOACs at the suggested dosages and very low rates of dose changes or interruptions over the post-index observation period. Patient-reported satisfaction with DOAC treatment was positive, with no patients reporting being dissatisfied at either the 3- or 6-month time point post-DOAC initiation.

The study found that the median time between stroke and DOAC initiation was 6 days, ranging from 0 days to 485 days post-stroke. The most common reason given for DOAC choice was clinical opinion. The majority of patients observed received the recommended doses of apixaban, edoxaban and rivaroxaban as per UK guidelines, with high levels of treatment persistence with the prescribed DOAC.

Regarding hospital care resource utilization, only 50% of patients had 1 or more outpatient clinic appointments recorded in the post-DOAC observation period. Adherence to DOACs as measured by the MMAS-8 was broadly consistent across the 3- and 6-month time points post-initiation.

The study's limitations include a smaller sample size than planned and potential impacts on observed resource utilization and subsequent patient follow-up due to the overlap in time between the study and the COVID-19 pandemic.

In conclusion, this study provides real-world evidence of the patient characteristics, management pathways, and patient-reported experiences of patients prescribed a direct FXa inhibitor DOAC after first stroke attributable to NVAF in UK clinical practice. The results provide clinicians with valuable insights into the experience of post-stroke patients with NVAF receiving direct FXa inhibitor DOACs treatment for secondary prevention of stroke.

Reviewer #3: <comments>

1. The authors have specifically indicated that a group of patients who have been diagnosed with non-valvular atrial fibrillation (NVAF) but are not receiving anticoagulation therapy are included in this cohort. Unfortunately, demographic information for this group of patients in the cohort is not available because it was not an intended outcome of this study. I recommend to include in the Limitations section that demographic information for this group of patients with known AF prior to stroke who were not anticoagulated, including those with a CHA2DS2 VASc Risk Score of 2 or more, was not available because it was not an intended outcome of this study. In addition, please note the following points: (1) In the Methods section, Group 1 is conoised of “patients with known AF prior to stroke who were not anticoagulated” and “those presenting with stroke who were found to have AF at the time of presentation or after”; and (2) The primary objective of this paper is stated as “to describe the demographics, clinical characteristics and medical history...”.

2. As the study flow diagram is not used, the author should describe the population of the statistics in more detail when describing each statistic. For instance, for the 86 patients, there is only a statement that “Of 86 patients with available information in their medical records.” If the patients are a group for whom medical records are available at 6 months, this should be stated as such. Similarly, I recommend that the author states what the population of 80 patients is in the HCRU and clinical assessments. I also suggest that the author may comment on the seven patients without medical records in the Limitations section.

3. The comment in the previous round means that the author is encouraged to state the rational reason for excluding dabigatran in Group 2, i.e. the reason for focusing on anti-FXa direct oral anticoagulants (DOACs). For instance, I suggest that the author may use descriptions such as “There are also limited data related to use of anti-FXa DOACs in routine clinical practice” from the protocol paper. Similarly, as with the protocol paper, I propose that DOACs be described as anti-FXa DOACs.

4. PACI, LACI, POCI, and TACI are not common abbreviations for non-neurologists, and as they only appear once in this paper, it is considered to be unnecessary to include as the abbreviations. Please consider deleting them.</comments>

7. PLOS authors have the option to publish the peer review history of their article (what does this mean? ). If published, this will include your full peer review and any attached files.

**Do you want your identity to be public for this peer review?** For information about this choice, including consent withdrawal, please see our Privacy Policy .

Reviewer #1: **Yes: ** Dimitrios Afendoulis

Reviewer #2: **Yes: ** Maha Ahmed

Reviewer #3: No

---

## [Author Response · Author response to Decision Letter 2]

24 Feb 2025

Response to reviewer comments R1:

We thank the reviewers for kindly spending the time to review this manuscript and providing their input. Below we have addressed the reviewers’ comments (Italicized below) and numbered them. Within the ‘marked’ copy of the revised manuscript, these changes can be found as tracked changes, in addition to being highlighted in yellow and with associated comments that contain the relevant comment number within this document.

Reviewer 1:

Comment

Very well presented manuscript.

Given the inclusion of the review comments it is suitable for publication

Response

We thank the reviewer for their positive comments.

Reviewer 2:

Comment

This study investigates the use of direct oral anticoagulants (DOACs) in adult patients with non-valvular atrial fibrillation (NVAF) who had a first ischaemic stroke. The study included 234 patients in Group 1 and 93 in Group 2. Group 2 patients were initiated on apixaban, edoxaban, or rivaroxaban post-first stroke. The primary objective was to describe patients' demographics, clinical characteristics, and medical history, stratified by the anticoagulant prescribed. The secondary objectives were to describe patient management pathways, hospital resource use, clinical assessments associated with DOAC treatment, and patient-reported satisfaction and experience of DOAC treatment.

The results demonstrate high levels of adherence, persistence, and treatment satisfaction in the 6 months post-initiation of DOAC after first stroke attributable to NVAF. Patients were largely initiated on DOACs in compliance with guidelines, with most patients prescribed DOACs at the suggested dosages and very low rates of dose changes or interruptions over the post-index observation period. Patient-reported satisfaction with DOAC treatment was positive, with no patients reporting being dissatisfied at either the 3- or 6-month time point post-DOAC initiation.

The study found that the median time between stroke and DOAC initiation was 6 days, ranging from 0 days to 485 days post-stroke. The most common reason given for DOAC choice was clinical opinion. The majority of patients observed received the recommended doses of apixaban, edoxaban and rivaroxaban as per UK guidelines, with high levels of treatment persistence with the prescribed DOAC.

Regarding hospital care resource utilization, only 50% of patients had 1 or more outpatient clinic appointments recorded in the post-DOAC observation period. Adherence to DOACs as measured by the MMAS-8 was broadly consistent across the 3- and 6-month time points post-initiation.

The study's limitations include a smaller sample size than planned and potential impacts on observed resource utilization and subsequent patient follow-up due to the overlap in time between the study and the COVID-19 pandemic.

In conclusion, this study provides real-world evidence of the patient characteristics, management pathways, and patient-reported experiences of patients prescribed a direct FXa inhibitor DOAC after first stroke attributable to NVAF in UK clinical practice. The results provide clinicians with valuable insights into the experience of post-stroke patients with NVAF receiving direct FXa inhibitor DOACs treatment for secondary prevention of stroke.

Response

We thank reviewer 2 for their comments.

Reviewer 3:

Comment

1. The authors have specifically indicated that a group of patients who have been diagnosed with non-valvular atrial fibrillation (NVAF) but are not receiving anticoagulation therapy are included in this cohort. Unfortunately, demographic information for this group of patients in the cohort is not available because it was not an intended outcome of this study. I recommend to include in the Limitations section that demographic information for this group of patients with known AF prior to stroke who were not anticoagulated, including those with a CHA2DS2 VASc Risk Score of 2 or more, was not available because it was not an intended outcome of this study. In addition, please note the following points: (1) In the Methods section, Group 1 is conoised of “patients with known AF prior to stroke who were not anticoagulated” and “those presenting with stroke who were found to have AF at the time of presentation or after”; and (2) The primary objective of this paper is stated as “to describe the demographics, clinical characteristics and medical history...”.

Response

Regarding the demographic information for those patients diagnosed with NVAF but not receiving anticoagulation, the following text has been added to the limitations section to reflect that this subgroup was not specified as an outcome and that this may be of interest for future research (pg 15) as suggested:

As it was not a predefined outcome, demographics were not specifically provided for those patients in Group 1 who were diagnosed with NVAF but had not received anticoagulation therapy prior to first stroke, however future research should attend to these patients.

Comment

2. As the study flow diagram is not used, the author should describe the population of the statistics in more detail when describing each statistic. For instance, for the 86 patients, there is only a statement that “Of 86 patients with available information in their medical records.” If the patients are a group for whom medical records are available at 6 months, this should be stated as such. Similarly, I recommend that the author states what the population of 80 patients is in the HCRU and clinical assessments. I also suggest that the author may comment on the seven patients without medical records in the Limitations section.

Response

The text has been edited for greater clarity as highlighted below (pg 10):

“Of the patients in Group 2 with available information in their medical records (n=86/93), 99% (n=85/86) were still being treated with their first DOAC at the end of the study observation period; one patient was documented to have died at 3.8 months.”

“Of the 93 patients in Group 2 with relevant available data in their medical records (n=80/93), half (50%, n=40/80; n=13 had no data recorded) had at least one outpatient visit within the post-DOAC observation period related to AF or DOACs….”

The text has also been edited to refer to the limitation of being reliant on the completeness of medical records (pg 16):

“Real-world data from medical records and patient-reported measures rely on the completeness of medical records and/or the given answers; these were not queried or followed up to clarify inconsistencies or complete missing answers”

Comment

3. The comment in the previous round means that the author is encouraged to state the rational reason for excluding dabigatran in Group 2, i.e. the reason for focusing on anti-FXa direct oral anticoagulants (DOACs). For instance, I suggest that the author may use descriptions such as “There are also limited data related to use of anti-FXa DOACs in routine clinical practice” from the protocol paper. Similarly, as with the protocol paper, I propose that DOACs be described as anti-FXa DOACs.

Response

This has been addressed throughout the manuscript utilizing the term “FXa-inhibiting DOACs” in order to be more specific to the mechanism of the drugs. Though it should be noted as Group 1 data included a range of DOACs (not only FXa-inhibiting DOACs) that the text only refers to FXa-inhibiting DOAC(s) where pertinent.

Comment

4. PACI, LACI, POCI, and TACI are not common abbreviations for non-neurologists, and as they only appear once in this paper, it is considered to be unnecessary to include as the abbreviations. Please consider deleting them.

Response

These abbreviations are the ones recorded from the medical records; thus they have been kept in the manuscript to reflect this. However, to improve clarity for the broader readership of the journal, the abbreviations have been moved into parentheses and the full text versions moved into the main text.

“…documented as having had partial anterior circulation infarcts (PACI), 22% (n=26) as having lacunar infarcts (LACI), 12% (n=14) as having posterior circulation infarcts (POCI) and 8% (n=9) recorded total anterior circulation infarcts (TACI).“

References

1. Scridon A, Balan AI. Challenges of Anticoagulant Therapy in Atrial Fibrillation—Focus on Gastrointestinal Bleeding. Int J Mol Sci. 2023 Apr 7;24(8):6879.

2. Shen NN, Zhang C, Wang N, Wang JL, Gu ZC, Han H. Effectiveness and Safety of Under or Over-dosing of Direct Oral Anticoagulants in Atrial Fibrillation: A Systematic Review and Meta-analysis of 148909 Patients From 10 Real-World Studies. Front Pharmacol [Internet]. 2021 Mar 18 [cited 2024 Nov 13];12. Available from: https://www.frontiersin.org/journals/pharmacology/articles/10.3389/fphar.2021.645479/full

3. Bhat Y, Dixit A, Mistri A, Patel B, Quoraishi SH, Uprichard J. A mixed methodology, non-interventional study to evaluate the use of direct oral anticoagulants in UK clinical practice for patients with a first stroke associated with non-valvular atrial fibrillation: study protocol. BMC Neurology. 2019 Nov 29;19(1):306.

4. National Institute for Health and Care Excellence. Atrial fibrillation: diagnosis and management. Atrial fibrillation [Internet]. 2021; Available from: https://www.nice.org.uk/guidance/ng196

5. Stephenson JJ, Shinde MU, Kwong WJ, Fu AC, Tan H, Weintraub WS. Comparison of claims vs patient-reported adherence measures and associated outcomes among patients with nonvalvular atrial fibrillation using oral anticoagulant therapy. Patient Prefer Adherence. 2018 Jan 12;12:105–17.

---

## [Decision Letter · Decision Letter 2]

11 Mar 2025

Real-world management, resource use, patient-reported outcomes and adherence in patients receiving direct oral anticoagulants for first stroke attributed to non-valvular atrial fibrillation in secondary care: a UK mixed-methods observational study

PONE-D-24-30678R2

Dear Dr. Uprichard,

We’re pleased to inform you that your manuscript has been judged scientifically suitable for publication and will be formally accepted for publication once it meets all outstanding technical requirements.

Kind regards,

Dr Redoy Ranjan, MBBS, MRCSEd, Ch.M., MS (CV&TS), FACS

Academic Editor

PLOS ONE

Additional Editor Comments (optional):

Reviewers' comments:

Reviewer's Responses to Questions

**Comments to the Author**

1. If the authors have adequately addressed your comments raised in a previous round of review and you feel that this manuscript is now acceptable for publication, you may indicate that here to bypass the “Comments to the Author” section, enter your conflict of interest statement in the “Confidential to Editor” section, and submit your "Accept" recommendation.

Reviewer #3: All comments have been addressed

2. Is the manuscript technically sound, and do the data support the conclusions?

Reviewer #3: Yes

3. Has the statistical analysis been performed appropriately and rigorously? 

Reviewer #3: Yes

4. Have the authors made all data underlying the findings in their manuscript fully available?

Reviewer #3: Yes

5. Is the manuscript presented in an intelligible fashion and written in standard English?

Reviewer #3: Yes

6. Review Comments to the Author

Reviewer #3: The author responds appropriately to the reviewers' comments. This article is considered to be acceptable and be worth publishing.

7. PLOS authors have the option to publish the peer review history of their article (what does this mean? ). If published, this will include your full peer review and any attached files.

**Do you want your identity to be public for this peer review?** For information about this choice, including consent withdrawal, please see our Privacy Policy .

Reviewer #3: No

---

## [Editor Report · Acceptance letter]

PONE-D-24-30678R2

PLOS ONE

Dear Dr. Uprichard,

I'm pleased to inform you that your manuscript has been deemed suitable for publication in PLOS ONE. Congratulations! Your manuscript is now being handed over to our production team.

Kind regards,

on behalf of

Dr. Redoy Ranjan

Academic Editor

PLOS ONE